# A Bio-Inspired Decision-Making Method of UAV Swarm for Attack-Defense Confrontation via Multi-Agent Reinforcement Learning

**DOI:** 10.3390/biomimetics8020222

**Published:** 2023-05-25

**Authors:** Pei Chi, Jiahong Wei, Kun Wu, Bin Di, Yingxun Wang

**Affiliations:** 1Institute of Unmanned System, Beihang University, Beijing 100191, China; 2School of Automation Science and Electrical Engineering, Beihang University, Beijing 100191, China; 3Flying College, Beihang University, Beijing 100191, China; 4Defense Innovation Institute, Academy of Military Sciences, Beijing 100071, China

**Keywords:** unmanned aerial vehicle, swarm, decision making, confrontation, multi-agent reinforcement learning

## Abstract

The unmanned aerial vehicle (UAV) swarm is regarded as having a significant role in modern warfare. The demand for UAV swarms with the capability of attack-defense confrontation is urgent. The existing decision-making methods of UAV swarm confrontation, such as multi-agent reinforcement learning (MARL), suffer from an exponential increase in training time as the size of the swarm increases. Inspired by group hunting behavior in nature, this paper presents a new bio-inspired decision-making method for UAV swarms for attack-defense confrontation via MARL. Firstly, a UAV swarm decision-making framework for confrontation based on grouping mechanisms is established. Secondly, a bio-inspired action space is designed, and a dense reward is added to the reward function to accelerate the convergence speed of training. Finally, numerical experiments are conducted to evaluate the performance of our method. The experiment results show that the proposed method can be applied to a swarm of 12 UAVs, and when the maximum acceleration of the enemy UAV is within 2.5 times ours, the swarm can well intercept the enemy, and the success rate is above 91%.

## 1. Introduction

With the development and maturity of unmanned aerial vehicle (UAV) flight control technology, the platform performance and intelligence level of UAVs are constantly improving. Therefore, the UAV is widely used in the military field and has become more and more significant in modern warfare [1,2,3]. Through collaboration among UAVs, the UAV swarm consisting of multiple UAVs can overcome the limitations of a single UAV in perception and execution and complete complex tasks [4,5,6,7,8,9], such as dynamic task allocation, collaborative reconnaissance, and attack-defense confrontation. Among these tasks, the method for attack-defense confrontation is highly valued as an emerging military technique that requires that the UAV make proper decisions autonomously according to the situation. The need for a UAV swarm with high-level confrontation intelligence is urgent.

This paper focuses on the attack-defense confrontation of a UAV swarm. Generally, in an attack-defense confrontation, the UAV swarm competes against a certain number of enemies with a certain level of intelligence to maximize their respective benefits. The objective of the UAV swarm mainly consists of two parts: destroying the enemy in a limited amount of time and protecting the base from the enemy’s invasion. The existing decision-making methods for attack-defense confrontations include matrix game methods, differential game methods, and expert system methods. However, these methods require some level of simplification and have the shortcoming that they are only suitable for small-scale and static scenarios. When the size of the UAV swarm is large and the scenarios are dynamic, it is hard to establish and solve the model.

In recent years, decision-making methods based on multi-agent reinforcement learning (MARL) have drawn a lot of attention. UAVs in the swarm are regarded as agents, and the agents receive rewards and learn the strategy through their interactions with the environment. Compared with other methods like differential game methods and expert system methods, methods based on MARL care less about the model of the system and are easier to design. Therefore, methods based on MARL are widely used by many researchers to solve the confrontation problem of UAV swarms. Since the solution space of the swarm confrontation problem is large and it is hard to obtain an effective strategy using standard MARL methods, researchers developed many methods based on MARL to increase the success rate. In [10], a hierarchical MARL framework for UAV swarm confrontation is proposed. A set of high-level macro actions and low-level primitive actions are designed to reduce the action space explored by the agents and increase the convergence speed. The experiment results show that the proposed method improves the success rate from 57% to 91% in 10 vs. 10 scenarios. A rule-coupled method [11] is realized based on the multi-agent deep deterministic policy gradient (MADDPG) algorithm. The rules are summarized and refined to guide the training of the agents. Compared with the original MADDPG algorithm, the rule-coupled method can obtain a better strategy with a higher success rate and shorter task completion time. The experiment results demonstrate that the UAV’s confrontation ability has improved. An improved multi-agent proximal policy optimization algorithm is proposed in [12]. The improved method adopts a framework of a centralized critic network and a decentralized actor network, which outperforms the framework of centralized critic network and centralized actor network in training time. The constraints of the environment and UAV dynamics are considered, and the method can achieve cooperation among UAVs without communication. A simulation environment for UAV swarm confrontation is constructed in [13]. In the scenario where 5 UAVs combat 5 UAVs, the performance of the multi-agent soft actor critic (MASAC) method and the MADDPG method are compared. The results show that the MASAC method can obtain a higher success rate than the MADDPG method. The weighted mean effect of interactions between UAVs is considered, and a weighted mean field reinforcement learning method for UAV swarm confrontation is proposed [14]. The method simplifies the multi-agent problem to a two-agent problem and can be applied to a large-scale UAV swarm. In [15], scenario-transfer training methods and self-play training methods are proposed to deal with complex scenarios, and a 3 vs. 3 UAV combatant scenario is constructed. These training methods can train a new model of complex tasks based on the model trained from simple tasks and accelerate the convergence speed. An inheritance training method [16] based on the multi-agent proximal policy optimization method is developed to improve the generalization performance of the model. The idea of course learning is adopted in the method, and the results show that UAVs can search for and attack targets outside the training area. However, the above methods mainly focus on increasing the success rate under the condition that the swarm size is fixed and small. For traditional MARL methods, the strategy trained for a certain number of UAVs is no longer feasible for a UAV swarm of a different size. Thus, the strategy has to be retrained as the swarm size changes. Due to the increase in swarm size, the dimensions of the state space and action space increase, and the solution space becomes larger. As a result, the training time increases exponentially as the swarm size increases.

To address the above problems, we get inspiration from the hunting behavior of pack predators. In nature, instead of flocking disorderly, many predators hunt for their prey by forming small-scale groups and making decisions autonomously through several types of interactions with each other. Compared with a large group, it is easier for small groups to cooperate. Inspired by this phenomenon, we propose a bio-inspired decision-making method for UAV swarms for attack-defense confrontation via multi-agent reinforcement learning. The main contributions of this paper are as follows:This paper proposes a bio-inspired decision-making method for UAV swarms for attack-defense confrontation via MARL. Traditional MARL methods suffer from an exponential increase in training time as the swarm size increases. To overcome this problem, the main idea of our method is to make the strategy trained for a small-sized UAV group applicable to a large-scale UAV swarm. Inspired by the phenomenon that predators hunt for prey in small groups, we propose the grouping mechanism, which divides the swarm into two types of groups. Through the grouping mechanism, interference between groups is avoided, so the strategy trained for small groups can be applied to a large-scale swarm, and the scalability of the UAV swarm is increased;To prevent the problem that the strategy is stuck in a local optimum during training, a bio-inspired action space is designed. Inspired by group hunting behavior in nature, we abstracted six types of actions that have a clear interactive effect. Compared with standard action space, the bio-inspired action space improves the success rate of the confrontation. Furthermore, as it is hard for the strategy to converge under a sparse reward, we design four types of dense rewards evaluating the status of the mission to accelerate the convergence of the strategy. The results show that an effective strategy can be obtained after adding dense rewards;The numerical experiments are conducted to evaluate our method. The results show that our method can obtain effective strategies and take advantage of the UAV swarm. The success rate of the confrontation is increased, and the UAV swarm can intercept the enemy, which is faster than itself, through cooperation.

This paper is organized as follows: In Section 2, the attack-defense confrontation problem is formulated, and the preliminary steps are introduced. In Section 3, the decision-making method of the UAV swarm for attack-defense confrontation is introduced in detail, including the framework, the grouping mechanism, and the design of MARL. In Section 4, the experiment results are presented, and the performance of our method is evaluated. In Section 5, the contribution of this paper is summarized, and future work is presented.

## 2. Preliminaries

### 2.1. Attack-Defense Confrontation Problem

In this paper, the attack-defense confrontation problem can be formulated as follows: As Figure 1 shows, it is assumed that our base has detected an enemy UAV approaching. To protect our base, k UAVs are launched to intercept the enemy UAV. The objective of the enemy UAV is to approach our base while evading our UAVs. If our base is within the detection range of the enemy UAV, it is considered that our base is exposed, and the interception mission fails. Considering that the enemy UAV may take countermeasures such as radar and infrared countermeasures to defend itself, the attack from one UAV is not 100% effective. Therefore, in this paper, only if the enemy UAV is within the attack range of four of our UAVs at the same time, it is considered that our UAVs cooperate to launch a saturation attack. In this case, it is confidently believed that the enemy UAV is destroyed and the interception mission succeeds.

As Figure 2 shows, the success conditions of the interception mission are defined as follows:(1)$$\left\|p_{i}(t_{suc})-p_{enemy}(t_{suc})\right\|\leq \rho_{atk},\exists U=\{u_{1},u_{2},u_{3},u_{4}\}\subseteq \{1,2,\ldots ,k\},\forall i\in U$$
(2)$$\left\|p_{base}-p_{enemy}(t)\right\|\geq \rho_{det},\forall t<t_{suc}$$
(3)$$0\leq t_{suc}\leq t_{max}$$
where $p_{i}$ represents the position of the *i*-th UAV, $p_{enemy}$ represents the position of the enemy UAV, $\rho_{atk}$ represents the attack range of our UAVs, and U represents a set containing a certain 4 of k UAVs. Each element u in set U represents a UAV, $p_{base}$ represents the position of our base, $\rho_{det}$ represents the detection range of the enemy UAV. Equation (1) represents that the enemy UAV is within the attack range of 4 of our UAVs at $t_{suc}$. Equation (2) represents that our base is not exposed before $t_{suc}$. Equation (3) represents that our UAVs should accomplish the interception mission in a limited time $t_{max}$.

### 2.2. Dynamics Model of the UAV

The UAV is assumed to be a mass point in a two-dimensional plane. The dynamic model of our UAVs is expressed as follows:(4)$$\left\{\begin{array}{l} {\dot{p}}_{i}=v_{i}\\ {\dot{v}}_{i}=a_{i}-\lambda v_{i} \end{array}\right.$$
where ${\dot{p}}_{i}$ represents the derivative of $p_{i}$, i.e., the velocity of the *i*-th UAV, $v_{i}$ represents the velocity of the *i*-th UAV, ${\dot{v}}_{i}$ represents the derivative of $v_{i}$, i.e., the acceleration of the *i*-th UAV, $a_{i}$ represents the control input of the *i*-th UAV, and λ represents the linear drag coefficient of the UAV.

Limited by the performance of UAV, the magnitude of the velocity and acceleration of UAV should meet certain constraints:(5)$$\left\|a_{i}\right\|\leq a_{max}$$
(6)$$\left\|v_{i}\right\|\leq v_{max}=\frac{a_{max}}{\lambda }$$
where $v_{max}$ and $a_{max}$ are the velocity limit constant and the acceleration limit constant, respectively.

Similarly, the dynamics model of the enemy UAV is expressed as follows:(7)$$\left\{\begin{array}{l} {\dot{p}}_{enemy}=v_{enemy}\\ {\dot{v}}_{enemy}=a_{enemy}-\lambda v_{enemy} \end{array}\right.$$
where ${\dot{p}}_{enemy}$ represents the derivative of $p_{enemy}$, i.e., the velocity of the enemy UAV; $v_{enemy}$ represents the velocity of the enemy UAV; ${\dot{v}}_{enemy}$ represents the derivative of $v_{enemy}$, i.e., the acceleration of the enemy UAV; $a_{enemy}$ represents the control input of the enemy UAV; and λ represents the linear drag coefficient of the UAV.

The magnitude of the velocity and acceleration of the enemy UAV should also meet certain constraints:(8)$$\left\|a_{enemy}\right\|\leq a_{max}^{enemy}$$
(9)$$\left\|v_{enemy}\right\|\leq v_{max}^{enemy}=\frac{a_{max}^{enemy}}{\lambda }$$
where $v_{max}^{enemy}$ and $a_{max}^{enemy}$ are the velocity limit constant and the acceleration limit constant, respectively.

### 2.3. Movement Strategy of Enemy UAV

In an attack-defense confrontation problem, the objective of the enemy UAV is to approach our base as close as possible while keeping as far away as possible from our UAVs. To make the enemy UAV move autonomously, we design the enemy UAV’s movement strategy based on the artificial potential field method. The basic idea is to assume that the enemy UAV is subject to an attractive force generated by our base and repulsive forces generated by our UAVs. The enemy UAV moves in a certain direction according to the combined force.

The control input $a_{enemy}$ of the enemy UAV is expressed as follows:(10)$$a_{enemy}^{\prime }=f\left(p_{base},p_{enemy}\right)+\sum_{i=1}^{k}g\left(p_{i},p_{enemy}\right)$$
(11)$$a_{enemy}=\left\{\begin{array}{cc} a_{enemy}^{\prime } & ,\left\|a_{enemy}^{\prime }\right\|\leq a_{max}^{enemy}\\ a_{max}^{enemy}\frac{a_{enemy}^{\prime }}{\left\|a_{enemy}^{\prime }\right\|} & ,\left\|a_{enemy}^{\prime }\right\|>a_{max}^{enemy} \end{array}\right.$$
where $f\left(p_{base},p_{enemy}\right)$ represents the attractive force and $g\left(p_{i},p_{enemy}\right)$ represents the repulsive force. They can be calculated using the following formulas:(12)$$f\left(p_{base},p_{enemy}\right)=a_{max}^{enemy}\frac{p_{base}-p_{enemy}}{\left\|p_{base}-p_{enemy}\right\|}$$
(13)$$g\left(p_{i},p_{enemy}\right)=-e^{-{\left(\frac{\left\|p_{i}-p_{enemy}\right\|}{\sqrt{2}\rho_{det}}\right)}^{4}}a_{max}^{enemy}\frac{p_{i}-p_{enemy}}{\left\|p_{i}-p_{enemy}\right\|}$$

The magnitude of the attractive force is constant, so the enemy UAV will move towards our base even if it is far from it. When the enemy UAV is far from our UAV, it is not necessary to change the movement direction. Therefore, only if the distance between the enemy UAV and our UAV is smaller than $\rho_{det}$, the magnitude of the repulsive force will be large enough to affect the movement direction of the enemy UAV.

### 2.4. Multi-Agent Reinforcement Learning

Reinforcement learning (RL) is a method that enables an agent to learn the optimal behavior strategy through interactions with the environment and is suitable for solving decision-making problems.

Multi-agent reinforcement learning (MARL) is an extension of RL in multi-agent systems. Typically, MARL algorithms adopt a framework of centralized training and decentralized execution (CTDE) [17,18]. The CTDE framework of MARL is shown in Figure 3.

There are two types of neural networks in the CDTE framework: actor networks and critic networks. The input of the actor network is the local observation of agent *i* denoted by $o^{i}$, and the output of the actor network is the action for agent *i* to execute, denoted by $a^{i}$. The input of the critic network is the joint state $s=(o^{1},o^{2},\ldots ,o^{n})$ consisting of all local observations and the joint action $a_{t}=(a^{1},\ldots ,a^{n})$, and the output of the critic network is the state-action value. At time step *t*, every agent selects its action independently according to its actor network. After the joint action $a_{t}$ is executed, the joint state $s_{t}$ will be updated, and the reward $r(s_{t},a_{t})$ received by all agents will be used to train the actor and critic networks.

The critic network parameterized by ϕ is trained by minimizing
(14)$$L(\phi )={\left(Q_{\phi }(s_{t},a_{t})-y\right)}^{2}$$
where L(ϕ) represents the loss function of the critic network parameterized by ϕ, $s_{t}$ represents the joint state at time step *t*, $a_{t}$ represents the joint action at time step *t*, $Q_{\phi }(s_{t},a_{t})$ represents the output of the critic network, y represents the expected output of the critic network, and
(15)$$y=r(s_{t},a_{t})+E\left[\sum_{l=1}^{\infty }\gamma^{l}r(s_{t+l},a_{t+l})\right]$$
where $r(s_{t},a_{t})$ represents the reward for executing the action $a_{t}$ in the state $s_{t}$, γ is a discount coefficient.

The actor network parameterized by μ is updated according to
(16)$$\nabla_{\mu }J(\mu )=E\left[\nabla_{\mu }\log\pi (a_{t}^{i}\;|\;o_{t}^{i})(Q_{\phi }(s_{t},a_{t})-b(s_{t},a_{t}))\right]$$
where J(μ) represents the objective function of the actor network parameterized by μ; $\pi (a_{t}^{i}\;|\;o_{t}^{i})$ represents the output of the actor network which is the probability for agent *i* to execute the action $a_{t}^{i}$ with the local observation; and $o_{t}^{i}$, $b(s_{t},a_{t})$ represents the baseline of state-action value.

## 3. Methods

### 3.1. Framework

In an attack-defense confrontation problem, our UAVs should decide how to move to intercept the enemy UAV. Inspired by the predatory behavior of pack hunters in nature, we propose a bio-inspired decision-making method for UAV swarms for attack-defense confrontation. We divide our UAVs into attack groups and backup groups according to the grouping mechanism. The attack group directly engages with the enemy UAV and learns movement strategy via multi-agent reinforcement learning. Backup groups adjust their formation according to the position of the enemy UAV and are ready to engage. The framework of the decision-making method of the UAV swarm for attack-defense confrontation is shown in Figure 4.

### 3.2. Grouping Mechanism

Based on the dataset of observations of wolves hunting elk in Yellowstone National Park, MacDulty suggests that the relationship between hunting success and group sizes is nonlinear [19]. When the group size is small, hunting success increases as the group size increases. However, hunting success peaks at a small group size and levels off when the group size is beyond 4. The reason for this phenomenon is that individuals in a small group cooperate better and their abilities are fully exhibited, while in a large group, individuals interfere with each other and some individuals cannot contribute to the hunt.

Similarly, when the group size of the UAV swarm is large, our UAVs interfere with each other, making it difficult to intercept the enemy UAV. Therefore, as shown in Figure 5, our UAV swarm is divided into several groups, and the area is divided into several zones. Every group is composed of four UAVs and is distributed in different zones. If the enemy UAV enters a zone, the UAV group in the zone becomes the attack group, and other UAV groups become the backup groups.

The attack group intercepts the enemy via MARL, which is presented in detail in Section 3.3. If the enemy UAV moves to other zones, the UAV group stops pursuing to prevent interfering with other UAV groups.

The backup groups should adjust their positions dynamically according to the position of our base and the enemy UAV. As shown in Figure 6, assuming that the current position of our base $p_{base}=\left(x_{b},y_{b}\right)$, the current position of the enemy $p_{enemy}=\left(x_{e},y_{e}\right)$, the current position of the formation center of the UAV group $p_{center}=\left(x_{c},y_{c}\right)$. The expected position of the formation center of the UAV group $p_{c}^{e}=\left(x_{c}^{e},y_{c}^{e}\right)$ should be on the line between our base and the enemy UAV.

The expected position of the formation center of the UAV group $p_{c}^{e}$ can be expressed as follows:(17)$$x_{c}^{e}=x_{c}$$
(18)$$y_{c}^{e}=y_{b}+\frac{y_{e}-y_{b}}{x_{e}-x_{b}}\left(x_{c}-x_{b}\right)$$

We design a discrete-time proportional-derivative (PD) controller to control the movement of UAVs in the backup groups. The control input $a_{i}(t)$ at time t for the *i*-th UAV can be determined as follows:(19)$$e(t)=\left\|p_{c}^{e}(t)-p_{c}(t)\right\|$$
(20)$$\left\{\begin{array}{c} a_{i}(t)=\left(k_{p}e(t)+k_{d}\left(e(t)-e(t-T_{s})\right)/T_{s}\right)\left(p_{c}^{e}(t)-p_{c}(t)\right)/e(t)\\ \left\|a_{i}(t)\right\|\leq a_{max} \end{array}\right.$$
where $k_{p}=2.5$, $k_{d}=2.2$, and $T_{s}=0.2s$ are parameters in the PD controller.

### 3.3. Design of MARL

The attack group is trained to intercept the enemy UAV based on MARL. Therefore, the elements of MARL, including action space, state space, and reward function, should be designed, respectively.

#### 3.3.1. Bio-Inspired Action Space

Many predators in nature hunt in groups for prey that is faster or larger than themselves. Similarly, in an attack-defense confrontation, our UAVs are predators, and the enemy UAV is the prey. Inspired by the hunting behavior of herd predators in nature, a bio-inspired action space is proposed. The bio-inspired action space contains two types of interaction: interaction between enemy UAVs and our UAVs and interaction among our UAVs.

(1)Interaction between Enemy UAVs and Our UAVs

MacNulty summarized the ethogram of large-carnivore predatory behavior by observing wolves in Yellowstone National Park [20]. He proposed that predatory behavior can be divided into six phases: search, approach, watch, attack-group, attack-individual, and capture. This paper focuses on the three main phases of group hunting behavior: approach, watch, and attack-individual, and abstracts these three phases into three types of action.

Approach. As shown in Figure 7, when our UAV and the enemy UAV are far apart, our UAV takes approaching action to quickly decrease the distance to the enemy UAV for performing the interception mission.

The control input of the *i*-th UAV can be calculated as follows:(21)$$a_{i}=a_{max}\frac{p_{enemy}-p_{i}}{\left\|p_{enemy}-p_{i}\right\|}$$

Watch. As shown in Figure 8, when our UAV is not within the detection range of the enemy UAV, it takes watching action to keep its distance from the enemy UAV and avoid causing the enemy UAV to escape. During this phase, our UAVs encircle the enemy UAV in preparation for the next phase of the interception mission.

When our UAV takes action, it moves clockwise or counter-clockwise with the enemy UAV as the center of the circle. As shown in Figure 9, taking clockwise motion as an example, the control input of the *i*-th UAV can be calculated as follows:(22)$$v_{t}=(v_{i}-v_{enemy})e_{t}$$
(23)$$a_{r}=\frac{v_{t}^{2}}{\left\|p_{enemy}-p_{i}\right\|}$$
(24)$$\theta =\cos^{-1}\left(\frac{a_{r}}{a_{max}}\right),\;a_{r}<a_{max}$$
(25)$$a_{i}=\left\{\begin{array}{cc} R(\theta )\cdot a_{max}e_{r} & ,\;a_{r}<a_{max}\\ a_{max}e_{r} & ,\;a_{r}\geq a_{max} \end{array}\right.$$
where $v_{t}$ represents tangential velocity of our UAV relative to the enemy UAV; $e_{t}$ represents the unit vector perpendicular to the line from the position of our UAV to the position of the enemy UAV; $a_{r}$ represents centripetal acceleration corresponding to tangential velocity; θ represents the angle between the direction of the control input of our UAV and the direction of the line connecting the enemy UAV and our UAV; $R\left(\theta \right)$ represents rotation matrix; and $e_{r}$ represents the unit vector in the direction of the line from the position of our UAV to the position of the enemy UAV.

Similarly, the control input of counter-clockwise motion can be calculated as follows:(26)$$a_{i}=\left\{\begin{array}{cc} R(-\theta )\cdot a_{max}e_{r} & ,\;a_{r}<a_{max}\\ a_{max}e_{r} & ,\;a_{r}\geq a_{max} \end{array}\right.$$

Attack-individual. As shown in Figure 10, similar to the harassment of the wolf pack, our UAVs induce the enemy UAV to move in a certain direction by constantly alternating between attack and retreat. In the process, our UAVs shrink the size of the encirclement, eventually achieving the capture of the enemy UAV.

It is noted that the direction of the control input during our UAV’s attack and retreat is not along the direction of the line connecting our UAV and the enemy UAV but rather towards the predicted future position of the enemy UAV.

The control input of an attack can be calculated as follows:(27)$$a_{i}=a_{max}\frac{p_{enemy}^{\prime }-p_{i}}{\left\|p_{enemy}^{\prime }-p_{i}\right\|}$$

The control input of retreat can be calculated as follows:(28)$$a_{i}=-a_{max}\frac{p_{enemy}^{\prime }-p_{i}}{\left\|p_{enemy}^{\prime }-p_{i}\right\|}$$

$p_{enemy}^{\prime }$ in Equations (27) and (28) represents the predicted future position of the enemy UAV, which can be calculated as follows:(29)$$p_{enemy}^{\prime }=p_{enemy}+\lambda_{p}\left\|p_{i}-p_{enemy}\right\|v_{enemy}$$
where $\lambda_{d}$ represents the prediction coefficient. The larger the prediction coefficient, the more distant the predicted future position.

Additionally, it can be seen that the predicted future position is related to the speed of the enemy UAV and the distance between the enemy UAV and our UAV. This is because the greater the speed of the enemy UAV or the greater the distance between the enemy UAV and our UAV, the greater the offset required to intercept, and the greater the distance between the predicted future position and the current position.

(2)Interaction among Our UAVs

In this paper, interaction among our UAVs is abstracted into three types of action: separation, alignment, and cohesion.

Separation. As shown in Figure 11, our UAVs take separation actions to prevent collisions between each other.

The control input of the *i*-th UAV can be calculated as follows:(30)$$a_{i}=\sum_{j=1,j\neq i}^{k}w_{j}\frac{p_{i}-p_{j}}{\left\|p_{i}-p_{j}\right\|}$$
where $w_{j}$ denotes the weighting factor which can be calculated as follows:(31)$$w_{j}=a_{max}\frac{\frac{1}{\left\|p_{i}-p_{j}\right\|}}{\sum_{j=1,j\neq i}^{4}\frac{1}{\left\|p_{i}-p_{j}\right\|}}$$

Alignment. As shown in Figure 12, our UAVs take action to keep each other at a certain distance and achieve group movement.

The control input of the *i*-th UAV can be calculated as follows:(32)$$a_{i}=a_{max}\frac{v_{avg}}{\left\|v_{avg}\right\|}$$
where $v_{avg}$ denotes the average velocity of other UAVs, which can be calculated as follows:(33)$$v_{avg}=\frac{1}{3}\sum_{j=1,j\neq i}^{4}v_{j}$$

Cohesion. As shown in Figure 13, our UAVs take action to approach each other and facilitate mutual support.

The control input of the *i*-th UAV can be calculated as follows:(34)$$a_{i}=a_{max}\frac{p_{avg}-p_{i}}{\left\|p_{avg}-p_{i}\right\|}$$
where $p_{avg}$ denotes the average position of other UAVs which can be calculated as follows:(35)$$p_{avg}=\frac{1}{3}\sum_{j=1,j\neq i}^{4}p_{j}$$

(3)Action Space

The action space of our UAVs contains nine actions, including approach, watch (clockwise), watch (counter-clockwise), attack-individual (attack), attack-individual (retreat), separation, alignment, cohesion, and void. Each action corresponds to a control input, and the control input for void is 0.

#### 3.3.2. State Space

The local observation $o_{i}$ of the *i*-th UAV consists of information from three parts: the enemy UAV, our base, and other UAVs. Specifically, $o_{i}$ can be expressed as follows:(36)$$p_{enemy}^{rel}=p_{enemy}-p_{i}$$
(37)$$v_{enemy}^{rel}=v_{enemy}-v_{i}$$
(38)$$p_{base}^{rel}=p_{base}-p_{i}$$
(39)$$p_{j,i}^{rel}=p_{j}-p_{i}$$
(40)$$o_{i}=\{p_{enemy}^{rel},v_{enemy}^{rel},p_{base}^{rel},p_{1,i}^{rel},\ldots ,p_{i-1,i}^{rel},p_{i+1,i}^{rel},\ldots ,p_{4,i}^{rel}\}$$
where $p_{enemy}^{rel}$ and $v_{enemy}^{rel}$ represent the relative position and the relative velocity of the enemy UAV, respectively; $p_{base}^{vel}$ represents the relative position of our base; and $p_{j,i}^{rel}$ represents the relative position of the *j*-th UAV.

#### 3.3.3. Reward Function

In MARL, the score $score_{suc}$ is usually determined based on the success of the task, and it is used as a reward r for training.

However, the biggest problem with such a setup is that the rewards are too sparse. Especially when it is hard to accomplish the task, the agents cannot obtain the rewards in a short time, and it is difficult to evaluate the quality of the current strategy. The direction of updating the strategy shows randomness, causing the problem that the algorithm is difficult to converge. To solve this problem, this paper modifies the reward function by adding prior knowledge to the reward function and by evaluating the current status, adding a dense reward to induce the agents to update the strategy in the direction of the superior status.

Considering that our UAVs need to approach the enemy UAV at a certain distance to perform the interception mission, a status evaluation function $score_{dis}$ related to the distance to the enemy UAV is added, and it can be expressed as follows:(41)$$score_{dis}=\mathrm{LJ}(\left\|p_{enemy}-p_{i}\right\|)$$
(42)$$\mathrm{LJ}(x)=\left\{\begin{array}{cc} 4\left[{\left(\frac{1}{1+\left(2\sqrt{2}-1\right)x/\rho_{atk}}\right)}^{2}-{\left(\frac{1}{1+\left(2\sqrt{2}-1\right)x/\rho_{atk}}\right)}^{4}\right] & ,x>\rho_{atk}\\ 1 & ,x\leq \rho_{atk} \end{array}\right.$$

The function value remains constant when the distance is smaller than $\rho_{atk}$, and it decreases gradually to 0 as the distance increases. Furthermore, the functions are smooth, bounded, and differentiable in their domains, which facilitates the training of the neural network and avoids gradient explosion.

Additionally, to avoid the enemy UAV escaping in the opposite direction from our UAVs, our UAVs should be scattered around the enemy and intercept the enemy from different directions. So, a status evaluation function $score_{encircle}$ related to the dispersion of our UAVs is added, and it can be expressed as follows:(43)$$\sigma =\sqrt{\sum_{i=1}^{4}\frac{{\left(\theta_{i}-\bar{\theta }\right)}^{2}}{4}},\bar{\theta }=\frac{1}{2}\pi$$
(44)$$score_{encircle}=1-2\pi \cdot \frac{4\sigma }{\sqrt{3}}$$
where $\theta_{i}$ represents the angle between the line connecting the *i*-th UAV and the enemy and the line connecting its counter-clockwise neighboring UAV and the enemy, as shown in Figure 14, σ represents the standard deviation of the angles.

Meanwhile, since the main goal of the interception mission is to prevent the enemy from approaching our base, the closer the enemy is to our base, the greater the threat to our base. A status evaluation function $score_{base}$ related to the distance to our base is added, and it can be expressed as follows:(45)$$score_{base}=-\mathrm{LJ}(\left\|p_{enemy}-p_{base}\right\|)$$
(46)$$\mathrm{LJ}(x)=\left\{\begin{array}{cc} 4\left[{\left(\frac{1}{1+\left(2\sqrt{2}-1\right)x/\rho_{det}}\right)}^{2}-{\left(\frac{1}{1+\left(2\sqrt{2}-1\right)x/\rho_{det}}\right)}^{4}\right] & ,x>\rho_{det}\\ 1 & ,x\leq \rho_{det} \end{array}\right.$$

Additionally, in the early period of training, it is easy for the enemy to invade our base. To update the strategy of our UAVs for hindering the enemy, a time reward function $score_{time}$ is added and it can be expressed as follows:(47)$$score_{time}=\frac{t}{t_{max}}$$

Therefore, the modified reward function for training is expressed as follows:(48)$$r=\omega_{s}score_{suc}+\omega_{d}score_{dis}+\omega_{e}score_{encircle}+\omega_{b}score_{base}+\omega_{t}score_{time}$$
where $\omega_{s}=10$, $\omega_{d}=2$, $\omega_{e}=3$, $\omega_{b}=3$, and $\omega_{t}=1$ are weighting factors. The weight parameters in (48) were selected according to empiricism. The greater the contribution of the function to the intercept mission, the greater the weight parameter.

## 4. Numerical Experiments

In this section, the strategy of the attack group is trained, and the strategy is applied to a swarm of 12 UAVs according to the grouping mechanism. Numerical experiments with enemies with different maximum accelerations are executed to test the performance of our method.

### 4.1. Experiment Setup

The experiment environment is built using Unity’s ML-Agents Toolkit. As shown in Figure 15, the training environment is 100 m long and 100 m wide. The circle on the left represents our base. The four squares represent four UAVs of the attack group. The circle on the right represents the enemy UAV. Parameters of the environment are listed in Table 1. The training parameters of MARL are listed in Table 2.

As Figure 16 shows, 12 UAVs are divided into 3 groups, and the environment is 175 m long and 100 m wide.

### 4.2. Performance Analysis

To validate the bio-inspired action space in our method, the success rates of the method with bio-inspired action space and the original action space in the training process are compared. The original action space contains five actions: up, down, left, right, and void. The curves of the success rates are shown in Figure 17, and the final success rates after 45,000 episodes of training are listed in Table 3.

It can be seen that both curves converged after 45,000 episodes of training. The curve with bio-inspired action space grew slowly in the early period of training, but it grew rapidly after about 45,000 episodes, and the success rate eventually remained at 97%. The curve with original action space grew rapidly in the early period of training, but it grew slowly after 24,000 episodes, and the success rate eventually remained at 89%. It shows that the bio-inspired action space can avoid being stuck in a local optimum and increase the final success rate. Compared to the original action space, the bio-inspired action space contains more types of actions, resulting in a slow growth in success rates in the early period. However, these actions have a clear interactive effect on both our UAVs and the enemy UAV, which facilitates the update of the strategy in a better direction.

After the strategy of the attack group is obtained, the success rate of the attack group against enemies with different maximum accelerations is evaluated. The results are shown in Figure 18 and Table 4.

The strategy is applied to a swarm of 12 UAVs, and the success rate against enemies with different maximum accelerations is obtained. The results are shown in Figure 19 and Table 5.

It can be seen that the success rate decreases as the maximum acceleration of the enemy UAV increases. Compared to the success rate of the attack group, the success rate of the UAV swarm is higher. The success rate against enemies with 3 times the maximum acceleration of ours increased from 2% to 53%. It shows that the grouping mechanism of our method can take advantage of the UAV swarm and increase the success rate. When the enemy’s maximum acceleration is within 2.5 times ours, our UAV swarm can intercept the enemy well, and the success rate is 91%.

### 4.3. Demonstration of Attack-Defense Confrontation

In this subsection, the process of the interception mission performed by the attack group and the UAV swarm is recorded.

Figure 20 and Figure 21 show how the attack group intercepts an enemy UAV. The maximum acceleration of the enemy is 0.45 m·s^−2^, the maximum speed of the enemy is 1.5 m·s^−1^, the maximum acceleration of our UAVs is 0.3 m·s^−2^, and the maximum speed of our UAVs is 1.0 m·s^−1^.

When the episode begins, the attack group approaches the enemy UAV to perform the interception mission. At t = 12 s, the speed of our UAVs shows a large difference. The speed of UAV 1 and UAV 4 is about 0.9 m·s^−1^, faster than the speed of UAV 2 and UAV 3, which is about 0.75 m·s^−1^. Thus, our UAVs form a U-shaped formation, which is helpful to avoid the enemy escaping. At t = 26 s, the enemy is within the attack range of our 4 UAVs, and the interception mission is successful.

Figure 22 and Figure 23 show how the UAV swarm intercepts an enemy UAV. Twelve UAVs are divided into three groups. Group 1 consists of UAVs 1 to 4. Group 2 consists of UAVs 5 to 8. Group 3 consists of UAVs 9 to 12. The maximum acceleration of the enemy is 0.75 m·s^−2^, the maximum speed of the enemy is 2.5 m·s^−1^, the maximum acceleration of our UAVs is 0.3 m·s^−2^, and the maximum speed of our UAVs is 1.0 m·s^−1^.

When the episode begins, group 3 approaches the enemy UAV, and groups 1 and 2 adjust their positions in their zones. From t = 20.9 s to t = 36.6 s, the enemy UAV, with the advantage of higher performance, accelerates to a higher speed to avoid the interception, breaks through the defense line formed by group 3 and enters the zone of group 2. Group 2 forms a U-shaped formation at t = 41.9 s and eventually intercepts the enemy UAV at t = 47.1 s.

Although, in the above process, the enemy UAV broke through the defense line formed by Group 3, Group 3 still played the role of hindering the enemy UAV and bought enough time for Group 2 to dynamically adjust the position. As the enemy UAV entered the zone of Group 2, Group 2 had already adjusted to a suitable position. So, it was able for Group 2 to quickly form an interception formation and realize the interception of the enemy.

## 5. Conclusions

This paper proposes a decision-making method for UAV swarms for attack-defense confrontation via MARL. For traditional MARL methods, the training time increases exponentially as the swarm size increases. Inspired by the phenomenon that many predators in nature hunt in small groups, our method abstracts the grouping mechanism to fully utilize the capability of the UAV swarm and mitigate interference between UAVs. The confrontation strategy is first obtained by training a group of four UAVs. Then, according to the proposed grouping mechanism, we apply the strategy to a larger-scale swarm. Therefore, even if the swarm size increases, the training time remains the same. Furthermore, to prevent the strategy from being stuck in a local optimum during training, six types of actions that have a clear interactive effect are generalized from hunting behavior. Several experiments are conducted to evaluate the performance of our method. The results show that when the maximum acceleration of the enemy UAV is within 2.5 times ours, a swarm of 12 UAVs can intercept the enemy well, and the success rate is above 91%. In addition, the grouping mechanism can take advantage of the UAV swarm and increase the success rate. And the method with the bio-inspired action space has a higher success rate compared with the method with the standard action space.

In this work, it is assumed that all UAVs are restricted to a 2D plane and that the UAV can obtain information about other UAVs without delay. Current work has mainly validated the effectiveness of our method on a simplified model. For future work, we will use a more precise dynamics model of UAVs and consider more constraints. Additionally, our method will be applied in a real-world flight experiment to demonstrate its feasibility.

## Figures and Tables

**Figure 1 biomimetics-08-00222-f001:**
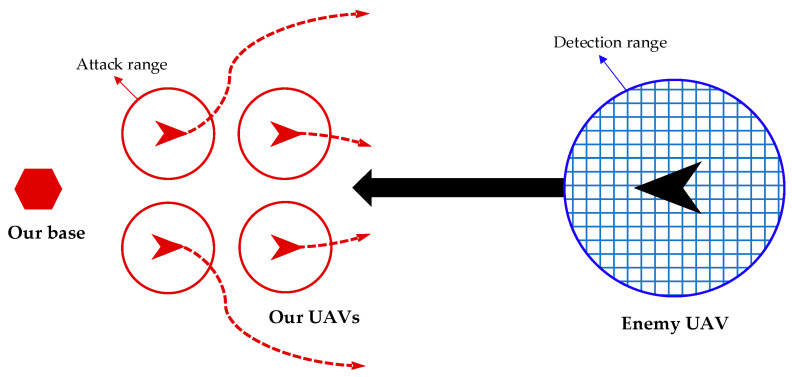
Attack-defense confrontation problem.

**Figure 2 biomimetics-08-00222-f002:**
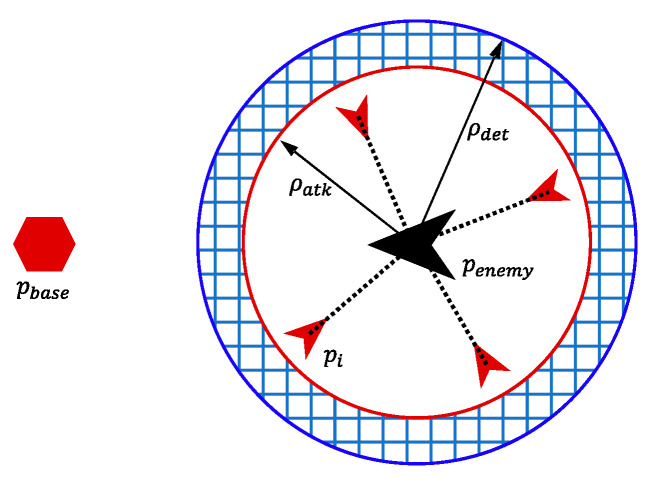
The success conditions of the intercept mission.

**Figure 3 biomimetics-08-00222-f003:**
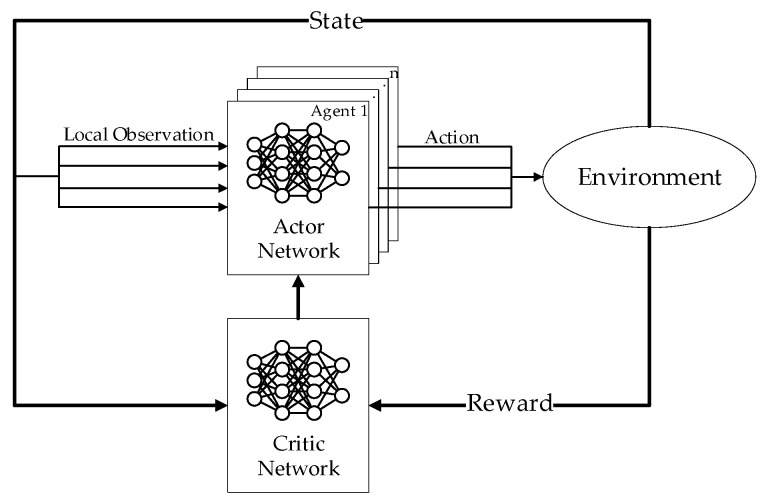
The CTDE framework of MARL.

**Figure 4 biomimetics-08-00222-f004:**
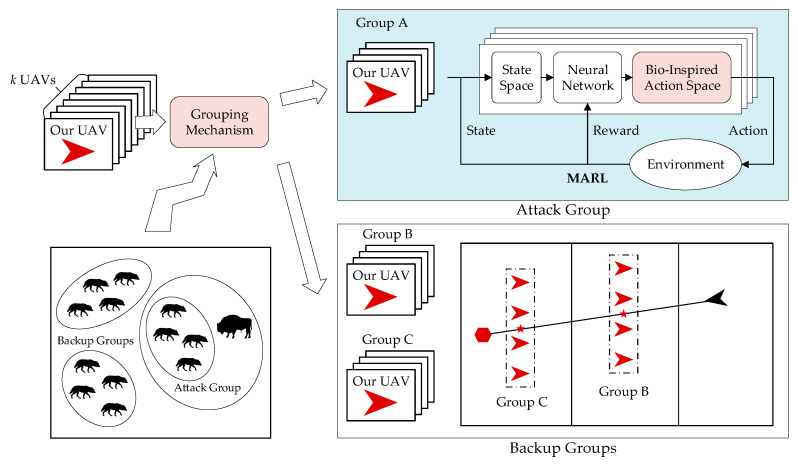
Framework of decision-making methods for UAV swarms for attack-defense confrontation.

**Figure 5 biomimetics-08-00222-f005:**
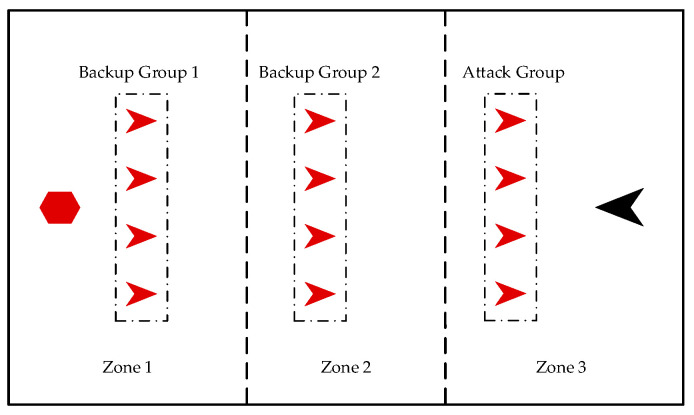
Grouping mechanism.

**Figure 6 biomimetics-08-00222-f006:**
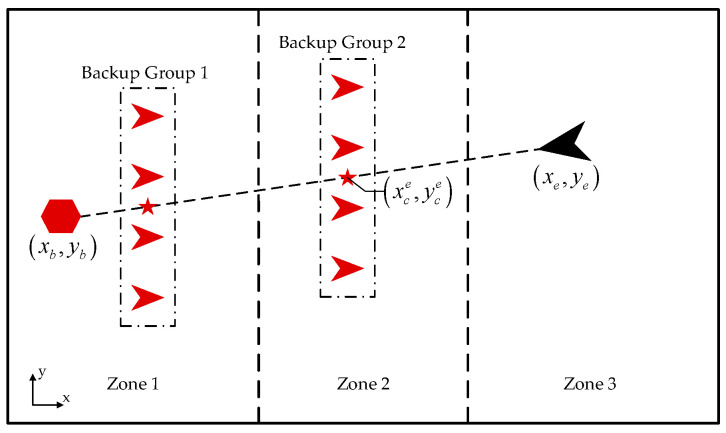
The movement strategy of backup groups.

**Figure 7 biomimetics-08-00222-f007:**
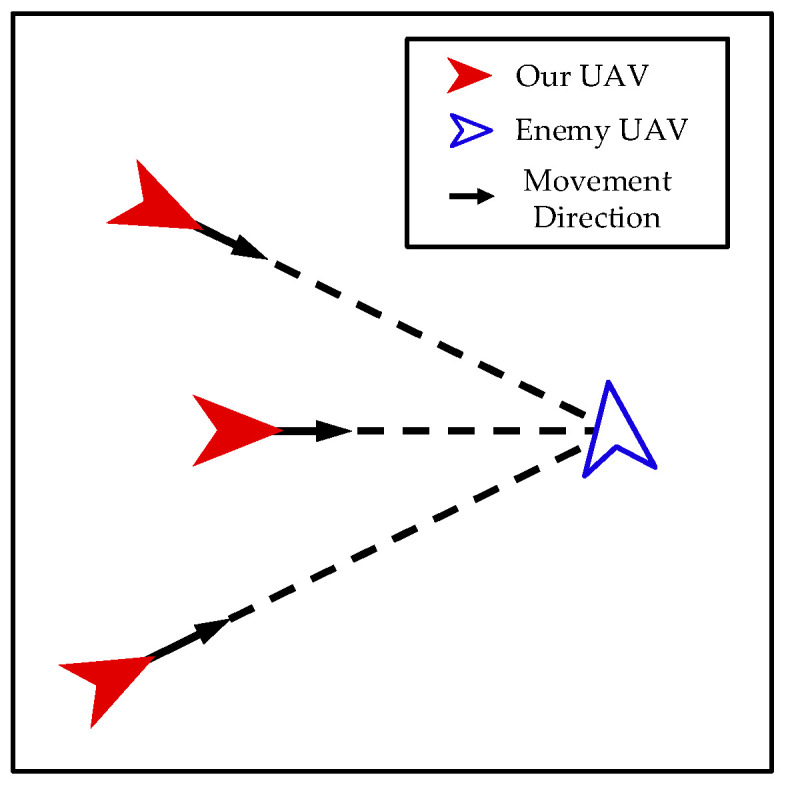
Approach.

**Figure 8 biomimetics-08-00222-f008:**
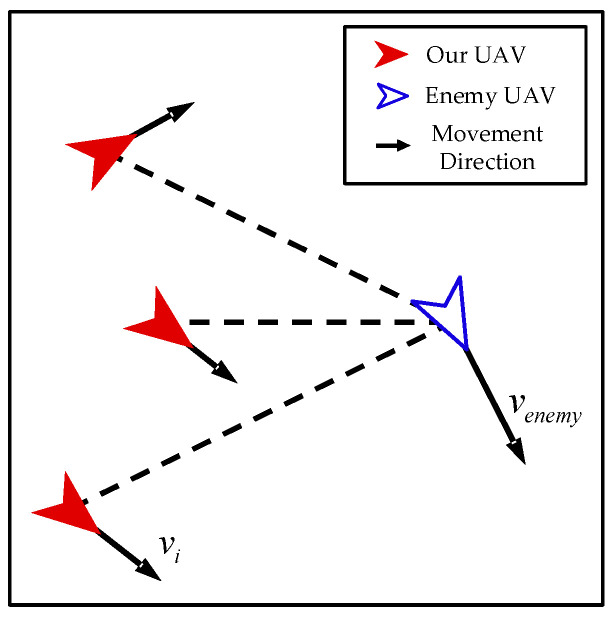
Watch.

**Figure 9 biomimetics-08-00222-f009:**
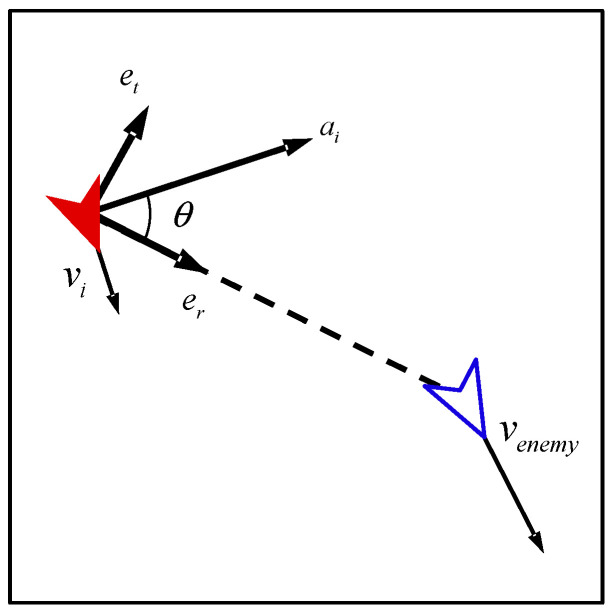
The control input of watch (clockwise).

**Figure 10 biomimetics-08-00222-f010:**
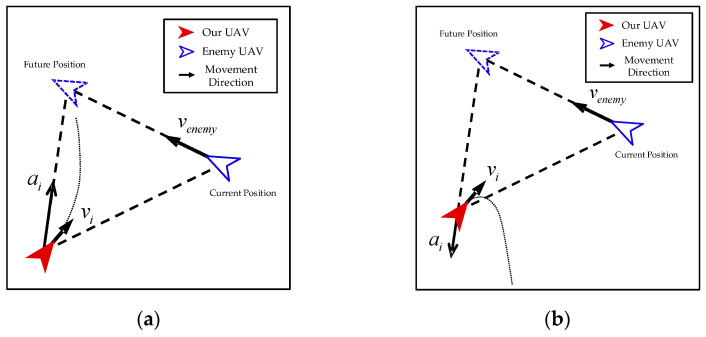
Attack-individual. (**a**) Attack; (**b**) retreat.

**Figure 11 biomimetics-08-00222-f011:**
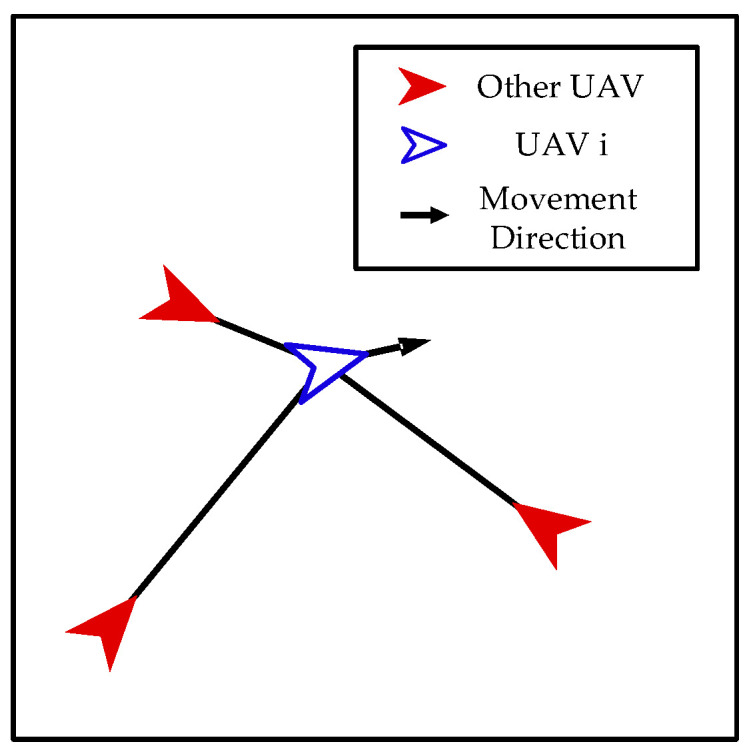
Separation.

**Figure 12 biomimetics-08-00222-f012:**
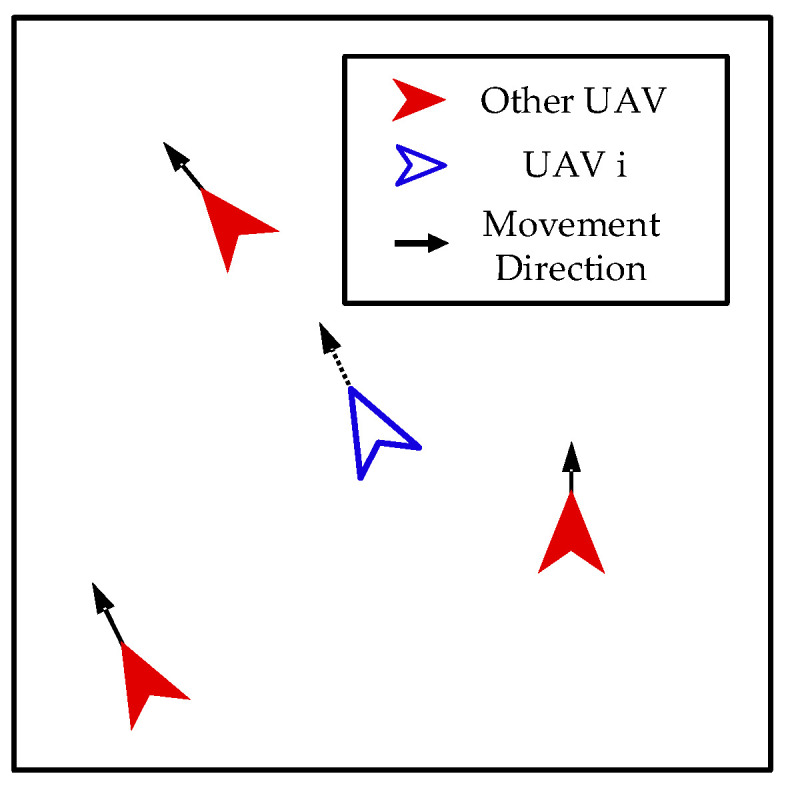
Alignment.

**Figure 13 biomimetics-08-00222-f013:**
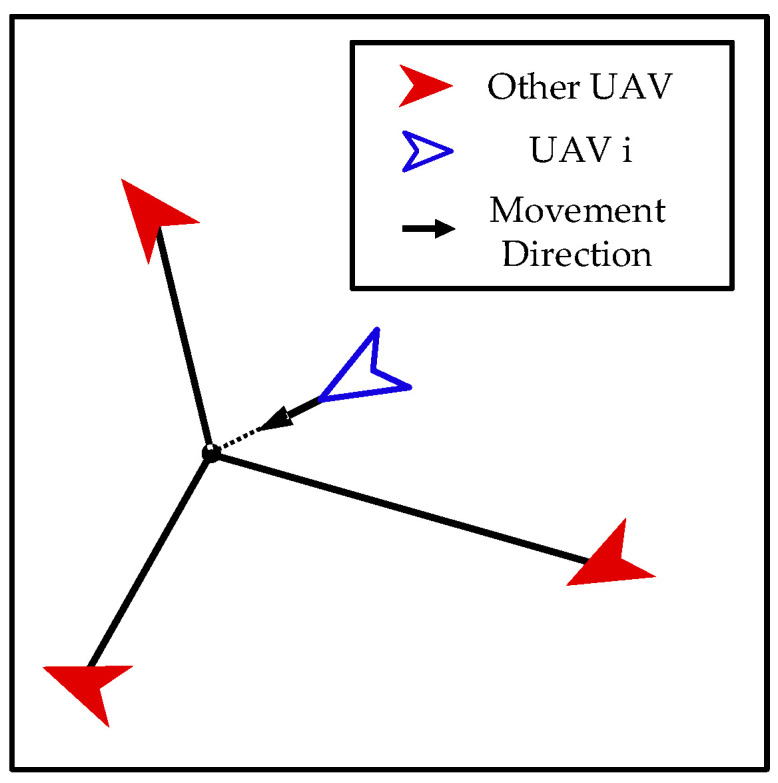
Cohesion.

**Figure 14 biomimetics-08-00222-f014:**
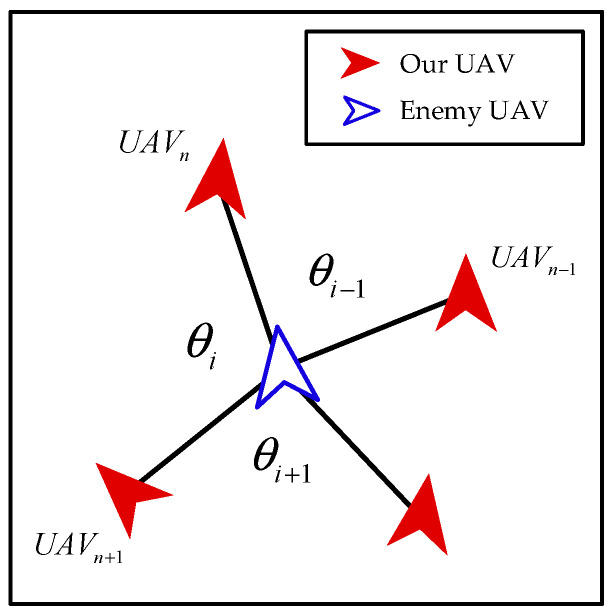
The definition of $\theta_{i}$.

**Figure 15 biomimetics-08-00222-f015:**
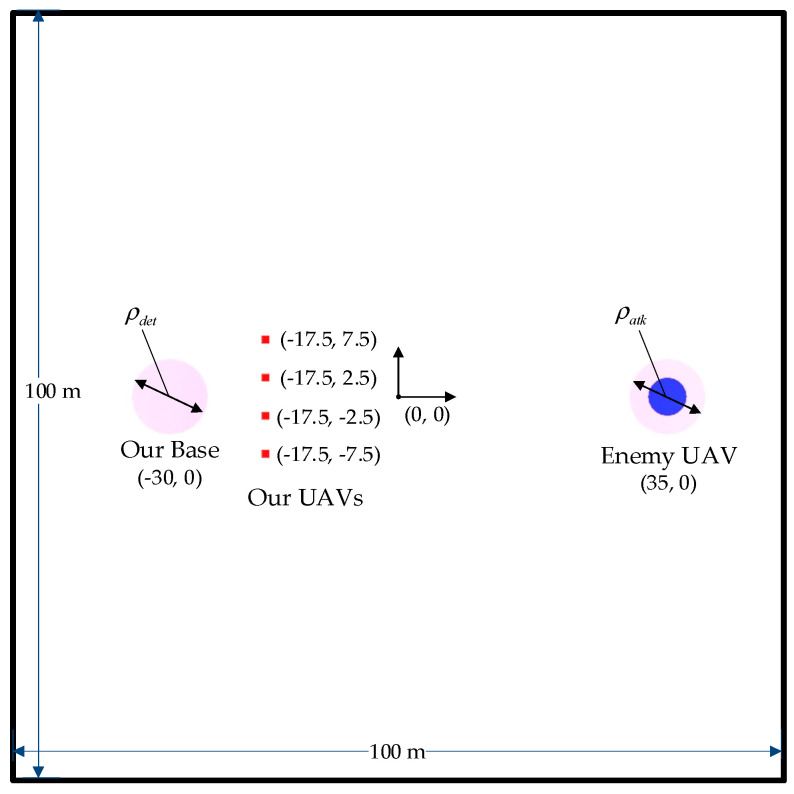
The initial state of the attack group in the experiment environment.

**Figure 16 biomimetics-08-00222-f016:**
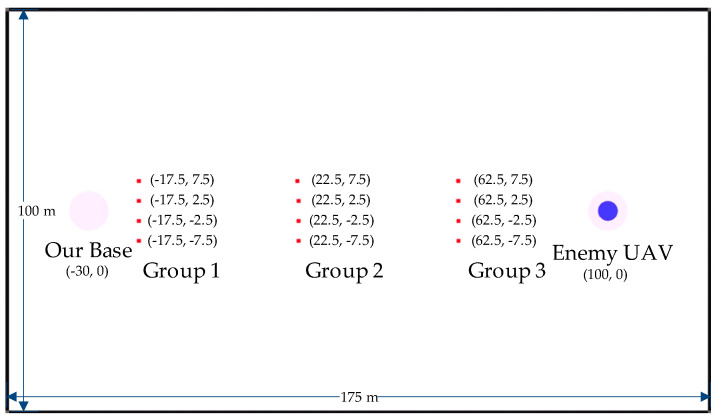
The initial state of the UAV swarm in the experiment environment.

**Figure 17 biomimetics-08-00222-f017:**
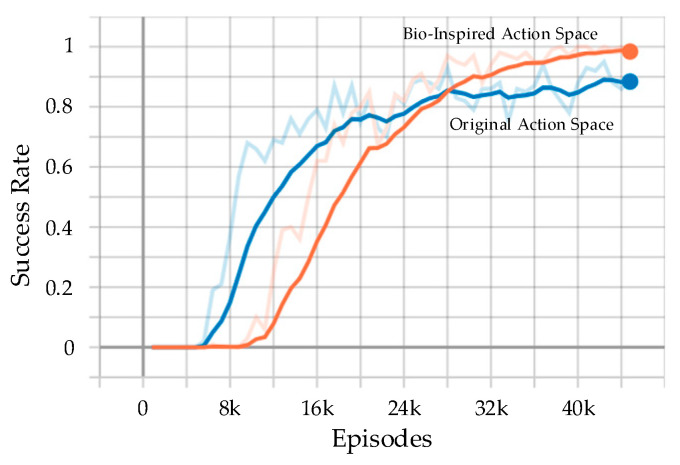
The curve of success rate per 100 episodes in the training process. Dim curves in the figure are the original curves of success rate, and bright curves are the smoothed ones using 1st-order low-pass-filter.

**Figure 18 biomimetics-08-00222-f018:**
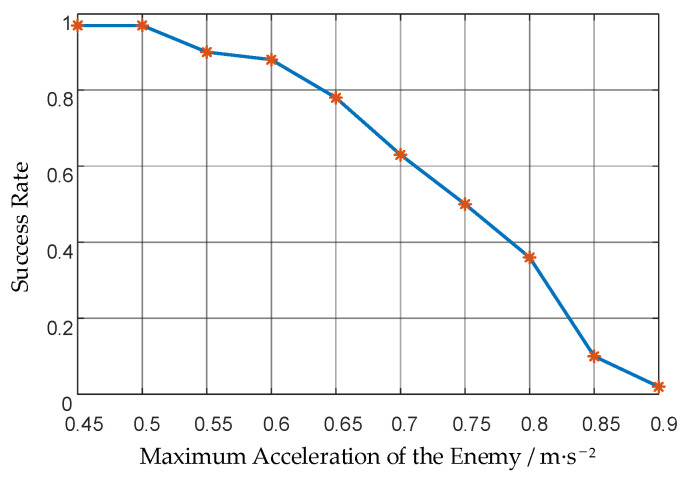
Success rates of the attack group against the enemy with different maximum accelerations.

**Figure 19 biomimetics-08-00222-f019:**
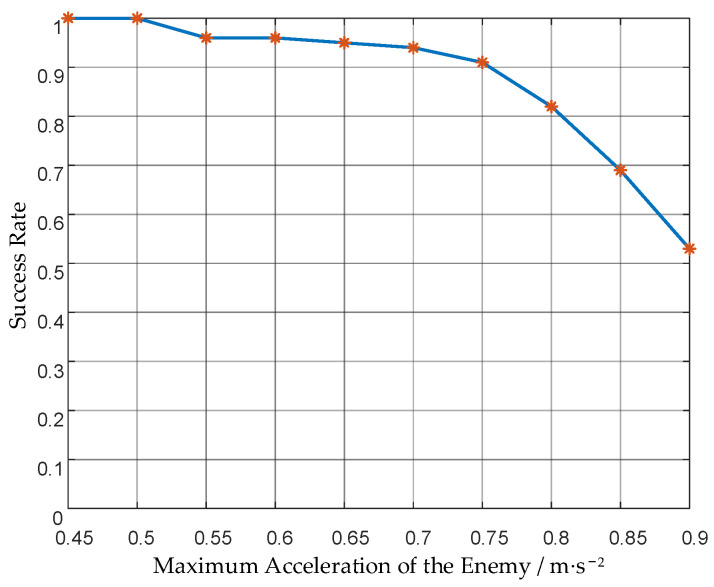
Success rates of the UAV swarm against the enemy with different maximum accelerations.

**Figure 20 biomimetics-08-00222-f020:**
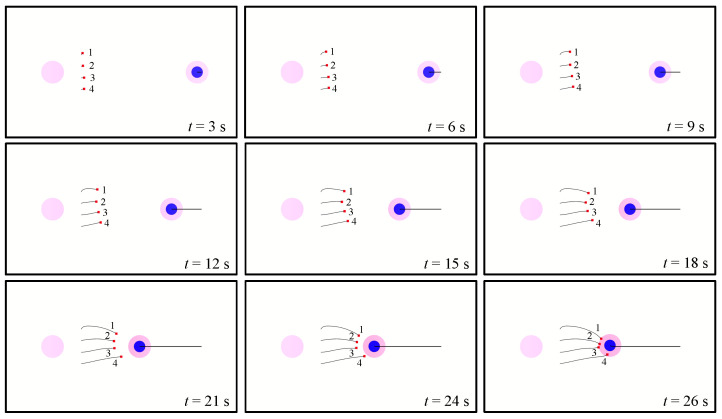
The trajectory of the attack group intercepting an enemy UAV. The number beside the square represents the number of the UAV. For the entire process of the mission, see Appendix A.

**Figure 21 biomimetics-08-00222-f021:**
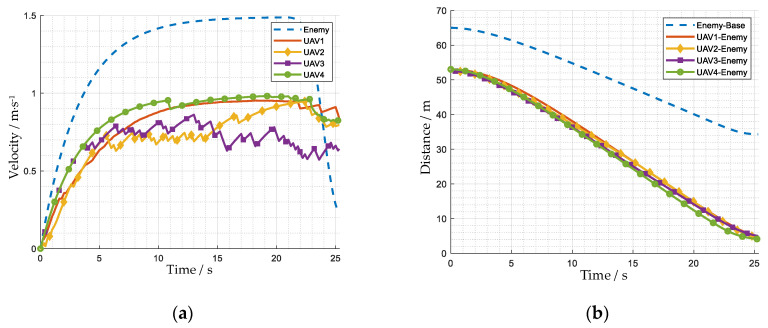
The state curves of the attack group and the enemy UAV. (**a**) Velocity; (**b**) distance.

**Figure 22 biomimetics-08-00222-f022:**
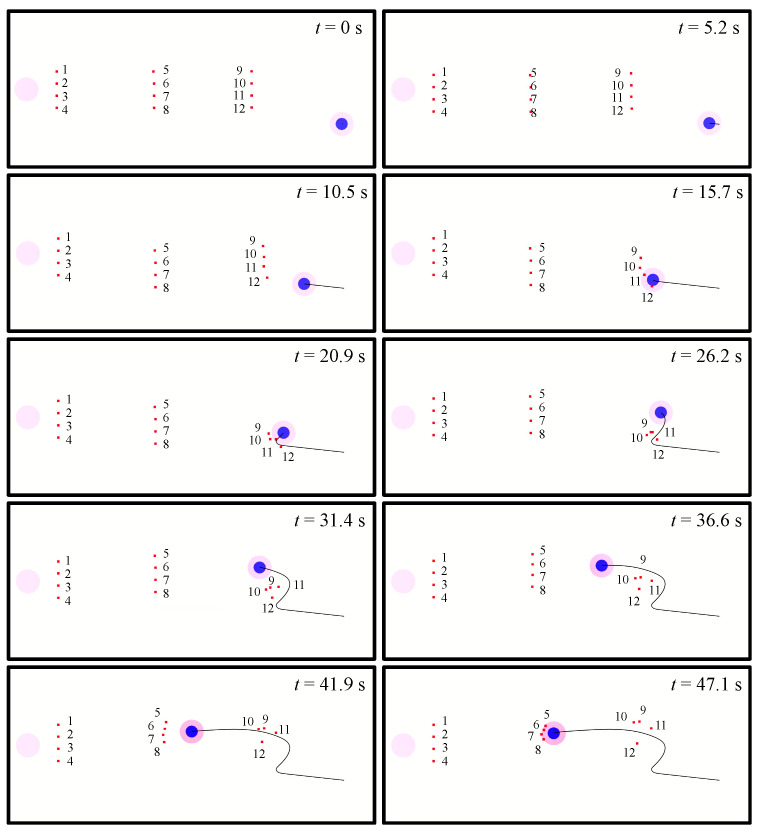
The trajectory of the UAV swarm intercepting an enemy UAV. The number beside the square represents the number of the UAV. For the entire process of the mission, see Appendix A.

**Figure 23 biomimetics-08-00222-f023:**
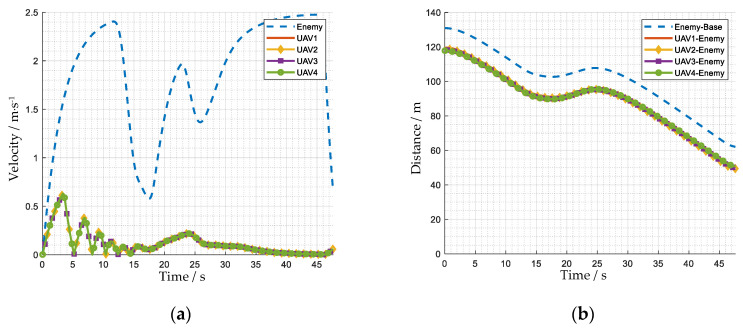
The state curves of the UAV swarm and the enemy UAV. (**a**) velocity of group 1; (**b**) distance between group 1 and the enemy; (**c**) velocity of group 2; (**d**) distance between group 2 and the enemy; (**e**) velocity of group 3; (**f**) distance between group 3 and the enemy.

**Table 1 biomimetics-08-00222-t001:** Parameters of the environment.

Parameter	Specification	Value
λ	Linear drag coefficient of the UAV	0.3 s^−1^
$a_{max}$	Maximum acceleration of our UAVs	0.3 m·s^−2^
$v_{max}$	Maximum speed of our UAVs	1.0 m·s^−1^
$\rho_{atk}$	Attack range of our UAVs	5.0 m
$a_{max}^{enemy}$	Maximum acceleration of the enemy UAV	0.45 m·s^−2^
$v_{max}^{enemy}$	Maximum speed of the enemy UAV	1.5 m·s^−1^
$\rho_{det}$	Detection range of the enemy UAV	5.0 m
$t_{max}$	Maximum time of the mission	500 s

**Table 2 biomimetics-08-00222-t002:** Training parameters of MARL.

Parameter	Value
Learning rate	0.00005
Batch size	1024
Buffer size	10,240
Discount factor	0.99
Hidden units	512
Fully connected layers	2

**Table 3 biomimetics-08-00222-t003:** Final success rates after 45,000 episodes of training.

Method	Final Success Rate
Original Action Space	89%
Bio-Inspired Action Space	97%

**Table 4 biomimetics-08-00222-t004:** Success rates of the attack group against the enemy with different maximum accelerations.

Maximum Acceleration of the Enemy/m·s^−2^	Maximum Acceleration of Our UAVs/m·s^−2^	Acceleration Ratio	Success Rate
0.45	0.3	1.5	97%
0.5		1.67	97%
0.55		1.83	90%
0.6		2	88%
0.65		2.17	78%
0.7		2.33	63%
0.75		2.5	50%
0.8		2.67	36%
0.85		2.83	10%
0.9		3	2%

**Table 5 biomimetics-08-00222-t005:** Success rates of the UAV swarm against the enemy with different maximum accelerations.

Maximum Acceleration of the Enemy/m·s^−2^	Maximum Acceleration of Our UAVs/m·s^−2^	Acceleration Ratio	Success Rate
0.45	0.3	1.5	100%
0.5		1.67	100%
0.55		1.83	96%
0.6		2	96%
0.65		2.17	95%
0.7		2.33	94%
0.75		2.5	91%
0.8		2.67	82%
0.85		2.83	69%
0.9		3	53%

## Data Availability

Not applicable.

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
