# Peer review of "A Bio-Inspired Decision-Making Method of UAV Swarm for Attack-Defense Confrontation via Multi-Agent Reinforcement Learning"

_biomimetics, 2023, doi:10.3390/biomimetics8020222_

Round 1

Reviewer 1 Report

In my viewpoint, the introduction has to be strengthened to more clearly state the purpose of the work and the features of the technique.
It's unclear how it operates as it stands.
The author should also make it apparent how this work advances state of the art, emphasizing its originality.

Each publication should make it apparent in the literature review what the suggested technique, innovation, and experimental outcomes are.
Highlight more clearly in a few words what general technical shortcomings in previous works were found to have prompted the development of the suggested strategy at the conclusion of related efforts.
You can use the following papers as references to clearly define the situation and the many potential solutions: https://www.sciencedirect.com/science/article/pii/S0167739X08000472 and https://www.sciencedirect.com/science/article/pii/S0957417421012598.

The section Conclusions should be developed further. The authors should first focus on their distinctive work and contributions and then support their conclusions with numerical data. The paper's shortcomings should then be explored. As a result, the future work of this study may be deduced.

Reviewer 2 Report

This paper presents a new bio-inspired decision-making method of a UAV swarm for attack-defense confrontation via multi-agent reinforcement learning (MARL) to solve the exponential increase in training time as the size of the swarm increases in traditional approaches. Numerical experiments conducted with a swarm of 12 UAVs evaluate the performance of the proposed method. 

The problem addressed is interesting, and the proposed solution seems promising. Nevertheless, the authors need to address the following issues.

a) The references redaction style "[X] ..." does not contribute to the readability of the paper.

b) No all variables in equation (1) are defined. The same in equations (14)-(16).

c) The description of models in equations (4) and (7) must be simplified. It is enough to say position, velocity, and acceleration. Why are the authors using "control volume" to refer to the control input? Any reference?

d) Under which constraints the PID gains were selected?

e) An animation of the proposed method must be provided.

f) Authors claim that as the size of the swarm increases in their proposed algorithm, the training time will not. How can the authors verify this claim?

Reviewer 3 Report

The paper “A Bio-Inspired Decision-Making Method of UAV Swarm for Attack-Defense Confrontation via Multi-Agent Reinforcement Learning” is devoted to developing UAV control strategy using reinforcement learning. The topic of the paper can be interesting for specialists in area of swarm system control. The paper is well-written, though some statements in the text are not clear and some additional comments are required. There are several suggestions and questions.

The attack-defense confrontation problem described in Section 2.1 is inspired by behavior of natural bio-systems. Though it is not technically clear how the enemy UAV attack can be intercepted by 4 our UAVs if they all are in attack-range. Some additional comments on this assumption will improve the understanding of the UAV system behavior.

Throughout the text the acceleration vector “a” is called “control volume”. It is suggested to replace it with “control acceleration”.

Why the authors use the model of friction acceleration as “-lambda*V”? More frequent model is quadratic dependence on the velocity: “-lambda*V^2*e_V”, where e_V is the unit vector along the velocity. Can the authors provide some reference with description of this model?

Why does the attractive force (12) and repulsive force (13) have different expressions? Shouldn't there be a minus sign in (13)? Some additional comments are required in the text.

The expression for the PD-controller in (20) does not include a term with the current velocity, which is not correct. How were the parameters selected? Can the authors comment on this?

The authors should explain the choice of the score functions presented in (42) and (46). Is there any meaning of the two terms? How were the weight parameters in (48) selected?

In Fig. 17 contains two bright curves and two dim curves. It seems that the bright curves are the results of averaging of the dim ones, though there is no comments on it in the text.

In simulation results section it will be also interesting to demonstrate the examples of the time-history plot of the current stage in the action space for the 4 UAVs. Which actions are of the most frequent use and which are probably redundant among the 9 actions?

Round 2

Reviewer 1 Report

The section Introduction should clarify better and provide concise information with regard to the problem definition and scope of the paper. The contribution summarization should be remarked on better. Moreover, the connection between the problem and the solution proposed is also not clearly pointed out. Emphasize the novelty introduced.

Highlight more clearly in a few lines what general technical shortcomings in previous works were found to have prompted the development of the suggested approach at the conclusion of related works. You should use the papers suggested in the previous revision as references to define the context clearly. Indeed, the depth and breadth of the literature survey are not enough, and it is necessary to supplement the advantages and weaknesses of directly related research on the basis of an extensive literature review.

It is necessary to strengthen the conclusion. Before providing quantitative evidence to support their conclusion, the authors should first highlight their original work and contributions. The limitations of the paper should then be discussed. Accordingly, the future work of this paper can be drawn.

Reviewer 2 Report

The paper's readability has been improved; the authors answered most of this reviewer's requests. However, this reviewer still feels that some parts of the paper need further clarification. 

1) The definition of set U needs to be clarified. Does each $u$ represent a UAV?

2) The definition for $\lambda$ at line 146 is not standard in the UAV literature. Could the authors give a reference or a more appropriate definition?

3) Although the authors provided animation for their results, it was done using a paid CODEC, so this reviewer could not visualize it.

4) The claim about not increasing the computing time due to swarm size needs to be supported with references.
